# Lock-in Amplifier-Based Impedance Detection of Tissue Type Using a Monopolar Injection Needle

**DOI:** 10.3390/s19214614

**Published:** 2019-10-23

**Authors:** Junsub Kim, Muhammad Aitzaz Abbasi, Taehee Kim, Ki Deok Park, Sungbo Cho

**Affiliations:** 1Gachon Advanced Institute for Health Science & Technology, Gachon University, Incheon 21999, Korea; askjs9020@naver.com; 2Department of Electronic Engineering, Gachon University, 1342 Seongnamdaero, Seongnam-si, Gyeonggi-do 13120, Korea; aitzazabbasi94@gmail.com; 3Department of Rehabilitation Medicine, Konkuk University Chungju Hospital, Chungju 27376, Korea; whitepoem37@naver.com; 4Department of Rehabilitation Medicine, Gil Medical Center, Gachon University College of Medicine, Incheon 21565, Korea; bduck@gachon.ac.kr

**Keywords:** bio-impedance, intra-articular injection therapy, lock-in amplifier, monopolar injection needle, porcine tissue

## Abstract

For successful intra-articular injection therapy, it is essential to accurately position the tip of the injection needle into the target joint area while administering the drug into the affected tissue. In this study, we investigated the feasibility of a monopolar injection needle and lock-in amplifier (LIA)-based impedance measurement system for detecting the tissue type where the needle tip is located. After positioning the monopolar injection needle tip into the dermis, hypodermis, or muscle layer of pork tissue, the electrical impedance was measured in the frequency range of 10 Hz to 10 kHz. We observed a difference in the results based on the tissue type where the needle was positioned (*p*-value < 0.01). Therefore, the monopolar injection needle with electrical impedance measurement can be used to improve intra-articular injection therapy through non-destructive and real-time monitoring of the needle position in the tissues.

## 1. Introduction

Intra-articular injection is an effective therapy for adhesive capsulitis and joint diseases such as rheumatoid arthritis and osteoarthritis [1,2,3]. It is important to know the location of the needle tip for accurate drug administration as the hypodermic needle is inserted into the affected joint or tissue where it delivers a dose of the drug. Incorrect location of the needle tip can diminish the effect of the drug or damage the tendons by causing rotator cuff tears [4]. The accuracy of the needle injection reaches 26.8%, even by a skilled operator under optimal conditions without an imaging guide [3]. Fluoroscopy (FL) or ultrasonography (US) can be helpful in determining the needle position in the tissue, and it has been seen that the accuracy of image-guided needle injection is better than blind injection [5,6,7,8,9]. However, FL-guided injection is accompanied with radiation exposure, and the visualization of soft tissues such as nerves or blood vessels is limited [10]. In contrast, although the US-guided injection is radiation-free, it has a low accuracy when the needle tip is inserted deep into tissue structures, due to its limited sensing distance and requirement of a skilled operator who can simultaneously control the injection needle as well as the US probe using both hands [11]. Therefore, novel alternative and supplemental methods are required to overcome the limitations of the traditional image-guided methods, as well as to improve the needle positioning and the intra-articular injection therapy.

One of the approaches to accurately position the needle tip in the target tissue is to electrically characterize the type of tissue into which the needle tip is being inserted by using microelectrodes that are integrated with the needle. Diverse needle types with microelectrodes have been utilized for procedures such as ablation of the heart or surrounding tissues [12], continuous glucose monitoring in vivo [13], and characterization of muscle tissues based on electromyography [14]. For the determination of the needle position in vivo, the tissue type can be characterized in a non-destructive and real-time manner using the needle integrated with microelectrodes and electrical impedance spectroscopy [15,16,17,18,19,20,21]. Electrical impedance is the ratio of the responsive potential to the applied weak alternating current, and the electrical impedance properties of the tissue are determined by its morphological structure and physiological condition, such as water contents, fat, and consecutive tissue [22]. The frequency-dependent electrical properties, conductivity and permittivity, of the biological tissues differ significantly from each other [23]. Previous literature reported the feasibility of the needle with different electrode configurations (monopolar, bipolar, or quadripolar) for the impedance characterization of the tissue types [15,16,19,24,25]. For operating the intra-articular injection therapy, the path of the needle insertion is determined to pass through the dermis, hypodermis, or muscle layer.

In this study, to improve the intra-articular injection therapy, we investigated the feasibility of using a disposable monopolar injection needle, which is insulated everywhere except at the needle tip, to facilitate identification of the type of pork tissue (dermis, hypodermis, or muscle) using electrical impedance measurement. Therefore, the monopolar injection needle has a higher current density and measurement sensitivity at the tip of the needle at the position of drug administration compared to the bi- or quadripolar electrode integrated needles. For the electrical impedance characterization of the tissues, we developed a virtual lock-in amplifier (LIA)-based impedance measurement system. LIA-based measurement is clearly efficient to remove the noise from the measured signal and to precisely extract the desired signal [26,27,28]. The developed impedance measurement system consisted of a data acquisition board (DAQ) board and LabVIEW program, which performed the function generator and virtual lock-in amplifier. Therefore, the system was simpler and cheaper than commercialized products. We performed LIA-based impedance measurement of the pork tissue where the tip of the monopolar injection needle was inserted using US as well as a developed LabVIEW programmed user interface. From the experimental results, we showed that application of the monopolar injection needle with LIA-based impedance measurement facilitates the accurate positioning of the needle tip for improved intra-articular injection therapy.

## 2. Methodology

### 2.1. Lock-in Amplifier-Based Impedance Measurement System

For the LIA-based impedance measurement system, a DAQ, which has a sampling rate of 102.4 kS/s, 0.8 Hz AC/DC coupling, 4-input/1-output channel for 24-bit analog–digital conversion (USB-4431, NI, Austin, TX, USA), was prepared. The mathematical functions for getting the LIA-based impedance data were implemented using LabVIEW programming. The AC signal for exciting the sample to be measured was generated from the digital-to-analog converter of the DAQ board and can be represented as follows:(1)$$V_{ex}=A_{ex}\sin(\omega t+\theta_{ex})$$
where *A_ex_* and *θ_ex_* are the amplitude and the phase of the AC signal, respectively, and *ω* is the angular frequency (= 2π*f*, *f*: Frequency).

When the generated *V_ex_* is applied on the sample in series to a current limiting resistor *R_cal_*, the divided potential *V_sam_* on the sample with unknown impedance (*Z_sam_*) is:(2)$$V_{sam}=\frac{Z_{sam}}{Z_{sam}+R_{cal}}\times V_{ex}=A_{sam}\sin(\omega t+\theta_{sam})$$
where *A_sam_* and *θ_sam_* are the amplitude and the phase of the measured potential of the sample, respectively.

If the measured potential *V_sam_* is mixed with the in-phase (*V_ex_in_*) or out-of-phase (*V_ex_out_*) exciting signal derived from the phase locked loop, phase shifter, and phase sensitive detector, the modulated signals (*V_mix_in_*, *V_mix_out_*) can be achieved as Equations (3) and (4).

(3)$$\begin{array}{l} V_{mix\_in}=V_{sam}\times V_{ex\_in}\\ \;\;\;\;\;\;\;\;\;\;\;\;\;\;=A_{sam}A_{ex}\sin(\omega t+\theta_{sam})\sin(\omega t+\theta_{ex})\\ \;\;\;\;\;\;\;\;\;\;\;\;\;\;=-\frac{1}{2}A_{sam}A_{ex}[\cos(2\omega t+\theta_{sam}+\theta_{ex})-\cos(\theta_{sam}-\theta_{ex})] \end{array}$$

(4)$$\begin{array}{l} V_{mix\_out}=V_{sam}\times V_{ex\_out}\\ \;\;\;\;\;\;\;\;\;\;\;\;\;\;\;\;=A_{sam}A_{ex}\sin(\omega t+\theta_{sam})\sin(\omega t+\theta_{ex}+\frac{\pi }{2})\\ \;\;\;\;\;\;\;\;\;\;\;\;\;\;\;\;=-\frac{1}{2}A_{sam}A_{ex}\sin(2\omega t+\theta_{sam}+\theta_{ex})-\frac{1}{2}A_{sam}A_{ex}\sin(\theta_{sam}-\theta_{ex}) \end{array}$$

After low pass filtering of the modulated signals, the in-phase (*V_r_*) and out-of-phase (*V_i_*) DC signals are acquired.

(5)$$V_{r}=\frac{1}{2}A_{sam}A_{ref}\cos(\theta_{sam}-\theta_{ex})$$

(6)$$V_{i}=-\frac{1}{2}A_{sam}A_{ref}\sin(\theta_{sam}-\theta_{ex})$$

Using the real and imaginary of measured potential, and Equation (2), the complex impedance value of the sample can be calculated.

### 2.2. Electrical Impedance Measurement of Porcine Tissues

The pork bellies from a 6-month-old mixed-breed pig were purchased from a butcher shop in the city of Seongnam, Korea and immediately taken to the laboratory over ice crystals to avoid the degradation of tissue. A piece of porcine skin composed of skin, dermis, hypodermis, and muscle was purchased and immersed in a saline (0.9% NaCl) solution to maintain the water content in tissue and the electrical conductivity of the tissue during the experiment. The porcine skin had the size of 150 mm (length) × 40 mm (width) × 50mm (height), and the weight of 0.3 kg. The experiment setup of the LIA-based impedance measurement system for detection of the tip position of the monopolar injection needle (25 gauge, Chalgren Enterprises, Inc., Gilroy, CA, USA) in the tissue layers with the Ag/AgCl counter electrode and saline injection with the syringe pump and ultrasound device is shown in Figure 1. The monopolar injection needle had an insulated outer wall with an outer diameter of 0.51 mm, an inner diameter of 0.26 mm, and a length of 50 mm. The monopolar injection needle was inserted into the pork tissue from the skin and positioned at a single tissue layer at a time with the help of US (SONON 300C, Healcerion Inc., Seoul, Korea). The electrical impedance spectrum of the pork tissue was measured in the frequency range of 10 Hz to 10 kHz, using the lock-in amplifier-based impedance measurement system and the Ag/AgCl counter electrode, which was placed on the skin of the pork tissue. The electrical impedance of the tissue was monitored, not only during the movement of the monopolar injection needle in the tissue layers, but also during injection of the saline (0.9% NaCl) solution used for arthrocentesis in a certain tissue layer. To avoid the effect of temperature on the impedance measurement of tissues, the temperature was kept constant (22 °C) during the experiment.

## 3. Results and Discussion

### 3.1. Accuracy of LIA-Based Impedance Measurement System

The accuracy of the developed LIA-based impedance measurement system was evaluated with respect to the resistors of 100 Ω to 1 MΩ or capacitors of 1 nF to 1 μF and compared to the commercialized product (PalmSens4, PalmSens Inc., Houten, Netherlands). Figure 2 shows the electrical impedance magnitude and the phase of the resistors (100 Ω to 1 MΩ) or capacitors (1 nF to 1 μF) measured by the developed LIA-based impedance measurement system (symbols) and the commercialized equipment (lines) in the frequency range of 10 Hz to 10 kHz. From the results, it was found that the LIA-based impedance measurement showed an accuracy comparable to the commercialized one.

### 3.2. Electrical Impedance Measurement of Pork Tissue

For the electrical impedance measurement of the pork tissue, the monopolar injection needle was positioned at a specific tissue layer using US. Figure 3 shows an ultrasound image of the pork tissue used for the experiment consisting of the epidermis (depth: 0.2 to 0.4 cm), dermis (0.4 to 1.1 cm), hypodermis (1.1 to 1.4 cm), and muscle (1.4 to 2.6 cm).

Figure 4 shows the impedance spectra measured in the frequency range of 10 Hz to 10 kHz after positioning the monopolar injection needle into different layers of the pork tissue or the saline (0.9% NaCl) solution. The electrical impedance of the monopolar injection needle in the tissue could be characterized by the capacitive electrode interfacial impedance at low frequencies below 100 Hz and the tissue impedance at higher frequencies. From the measured impedance spectrum of tissue normalized to the data of the saline (0.9% NaCl) solution, we observed a significant difference in the impedance value, based on the tissue type at high frequencies above 1 kHz (Figure 4c,d). The measured impedance was more affected by the electrical properties of the neighbored tissue as the needle tip was located close to the neighboring tissue. The impedance measurement with the monopolar injection needle could not separate the impedance of each tissue layer from the total measured impedance data. Nevertheless, it was able to distinguish the impedance of the dermis, hypodermis, or muscle using the LIA-based impedance measurement with the monopolar injection needle.

### 3.3. Impedance Monitoring of the Monopolar Injection Needle Position

Based on the impedance spectrum measured at different tissue layers, the frequency of 2.15 kHz was selected for the impedance monitoring of the needle position. Figure 5a shows the real and imaginary part of the impedance at 2.15 kHz, recorded when the needle tip was located at different depths in each tissue layer. The real part of the impedance at 2.15 kHz was significantly different between the tissue layers, as shown in Figure 5b (*p*-value < 0.001, evaluated by Student’s *t*-test). From the monitoring of the real part of the impedance at the selected frequency of 2.15 kHz, when the monopolar injection needle was moved into the tissue layers, as shown in Figure 5c, it was found that the position of the needle tip could be successfully detected. The variations in the impedance data caused by needle movement occurred and increased especially at the boundary of the tissue layers. The variations of the impedance measurement during the needle movement were thought to be caused by the different depths in the tissue or the relative distance between the working and counter electrodes (see Figure 4). However, the impedance data measured in different tissues could be significantly separated from each other. While the needle tip was placed in the dermis, the value corresponding to the real part of impedance was highest compared to the values of other tissues. However, the value was lowest when the needle tip was positioned in the muscle tissue. Additionally, the real part of the impedance of the tissue layer was measured before and after the saline (0.9% NaCl) injection process and shown in Figure 5d. After the saline injection in each tissue layer, the measured real part of the impedance decreased due to the higher conductivity of the saline than one of the tissues, and then gradually increased according to the recovery of the water content. The traced impedance data could be clearly distinguished according to the position of the needle at different tissue types.

The water content varied with not only tissue type, but also with race, age, and health condition [29]. From the impedance measurement of wholly different tissue samples, it was found that the impedance of the dermis, hypodermis, or muscle could be distinguished. The relative difference in the impedance data of the dermis, hypodermis, or muscle agreed with previous studies [15,16,21]. By increasing the frequency to higher than 10 kHz, we were able to additionally analyze the impedance characteristics of the intracellular area; however, we then needed to compensate for the stray currents [30]. The results proved that the simple DAQ board using the LabVIEW program could successfully distinguish the impedance of tissue type in the frequency range of 10 Hz to 10 kHz. Consequently, the position of the monopolar injection needle tip in the tissue could be accurately and continuously detected by the electrical impedance monitoring. It is expected that this technique can be applied to animals or humans for positioning the injection needle into the joint cavity filled with synovial fluid, which has similar conductivity to the saline solution, and for better intra-articular injection therapy.

## 4. Conclusion

We investigated the feasibility of the LIA-based impedance measurement with a monopolar injection needle for monitoring the position of the needle tip in different tissue layers to improve intra-articular injection therapy. The developed LIA-based impedance measurement system showed a measurement accuracy for the resistors (100 Ω to 1 MΩ) or capacitors (1 nF to 1 μF) in the frequency range of 10 Hz to 10 kHz, comparable to the commercialized product. The electrical impedance of the tissue was measured by positioning the monopolar injection needle in the pork tissue with the aid of US. The impedance data of the tissue layer measured at 2.15 kHz were found to be significantly different from each other (*p*-value < 0.001, evaluated by Student’s *t*-test). The experimental results proved that the position of the needle tip in the tissue layer could be accurately and continuously detected by electrical impedance monitoring. Therefore, we expect that the impedance measurement-based needle positioning can be utilized to improve intra-articular injection therapy.

## Figures and Tables

**Figure 1 sensors-19-04614-f001:**
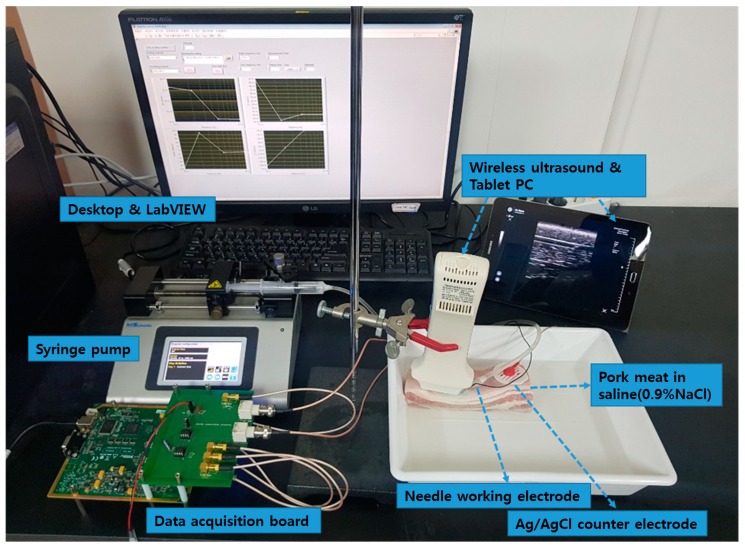
Experimental setup of the lock-in amplifier (LIA)-based impedance measurement system for detection of the tip position of the monopolar injection needle in the tissue layers with Ag/AgCl counter electrode and saline injection with the syringe pump and ultrasound (US) device.

**Figure 2 sensors-19-04614-f002:**
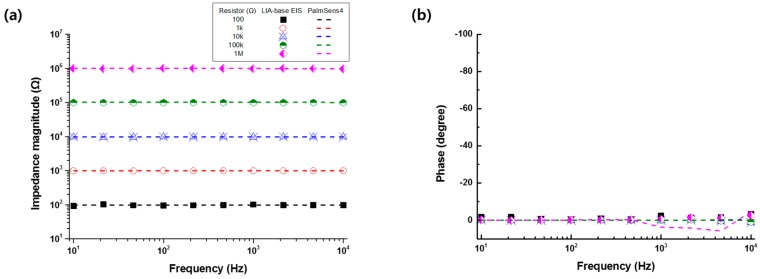
(**a**) Electrical impedance magnitude and (**b**) phase of the resistors (100 Ω to 1 MΩ) or the (**c**) impedance magnitude and (**d**) phase of capacitors (1 nF to 1 μF) measured by the developed impedance measurement system (symbols) and commercialized product (PalmSens4, lines) in the frequency range of 10 Hz to 10 kHz.

**Figure 3 sensors-19-04614-f003:**
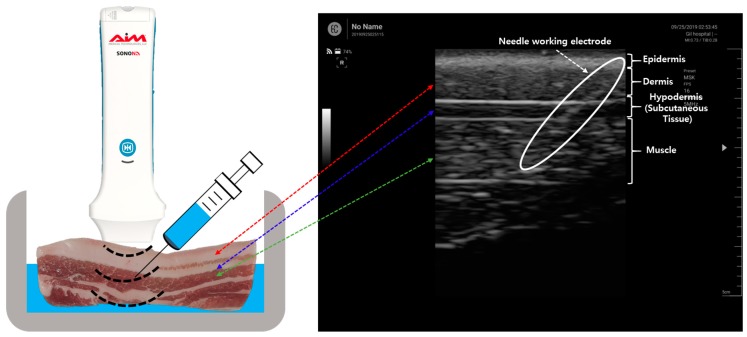
Ultrasound image of the needle insertion in the tissue layer.

**Figure 4 sensors-19-04614-f004:**
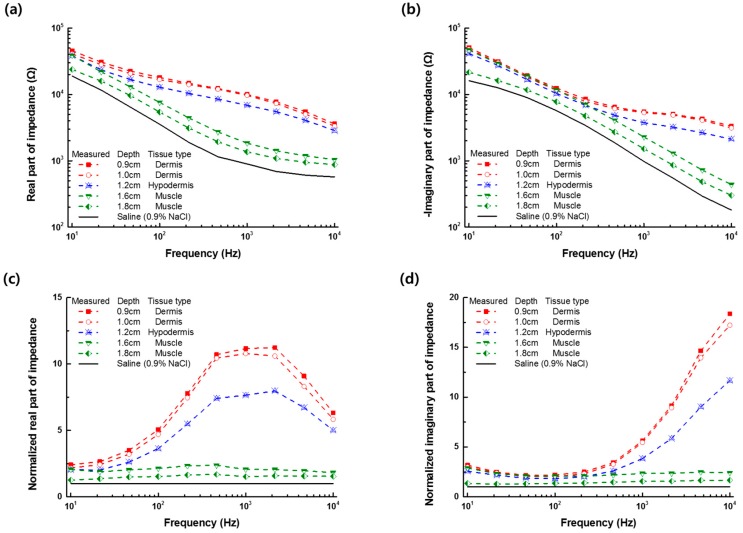
(**a**) Real and (**b**) imaginary part of the impedance spectra measured at different depths of the needle in the dermis, hypodermis, or muscle tissue; and normalized (**c**) real and (**d**) imaginary part of the impedance spectra to the data of the saline (0.9% NaCl) solution.

**Figure 5 sensors-19-04614-f005:**
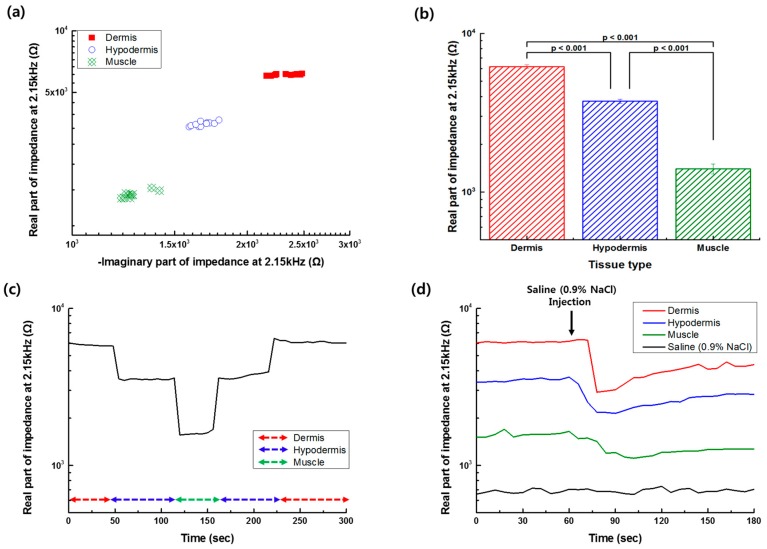
(**a**) Real and imaginary part of the impedance of the tissue layer measured at 2.15 kHz. (**b**) Difference of the average of the real part of the impedance at 2.15 kHz between the tissue layers (n = 3, bar: Standard error) evaluated by the Student’s *t*-test. (**c**) Monitoring of the real part of the impedance at 2.15 kHz, while the monopolar injection needle is moved to a different tissue layer. (**d**) Monitoring of the real part of impedance at 2.15 kHz during the saline injection process at different tissue layers.

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
