# Peer review of "Lock-in Amplifier-Based Impedance Detection of Tissue Type Using a Monopolar Injection Needle"

_sensors, 2019, doi:10.3390/s19214614_

Round 1

Reviewer 1 Report

Overall, this paper discusses the use of a monopolar injection needle to measure the impedance of different tissues. The impedance is measured with a lock-in-amplifier based impedance measurement system. It successfully distinguishes three different types of tissue and is suggested to improve intra-articular injection therapy by allowing accurate position of the needle tip in a target tissue. However, it does not state why this proposed method is an improvement and there is no supporting data for it. Previous works have successfully identified both tissue type and depth into tissue, but the paper does not indicate advantages and does not compare those results to existing data. 

Additional notes:

What is the benefit of the proposed method over existing works? The detection of different tissue types (muscle, fat, and nerve) has previously been accomplished by measuring impedance with either a monopolar or bipolar needle (from reference 15). Additionally, impedance was measured with a needle to determine both depth and type of tissue (reference 16 and 20) and there has been an in vivo study to distinguish different tissues by impedance with a needle (reference 19). A reasoning or advantage for using a lock-in amplifier system to measure tissue impedance isn’t stated, and the results of this paper are not compared to expected tissue impedances or existing impedance measurement system. This information would be helpful to indicate the reasoning for this work and show why it’s useful. Is the depth into tissue important for therapy, and is this impedance measurement system able to determine depth into tissue as in previous works? The authors state that “Properties of the needle such as dimension, configuration, material of the needle electrode, and measurement frequency need to be carefully selected as electrical impedance measurement of the biological tissue is affected by the capacitive electrode polarization impedance, which is influenced in turn by the surface states of the electrode.” However, the authors do not indicate properties of the needle used. It is also unclear if these effects are significant enough to cause a problem in determining the tissue type. For intra-articular injection therapy, what tissues do operators need to identify? Is it only the dermis, epidermis, and muscle? Or is nerve tissue and other tissues important? Figure 1 shows an illustration of the experimental setup. Including a picture may be useful to show where the needle was inserted, where the reference electrode was placed, and the size of the meat. What was the size or volume of the meat? The tissue is place in an electrolyte solution to maintain water content and conductivity. Does water content vary throughout the tissue? Would water content vary between patients? If so, would this cause any issues with correctly determining tissue type? Figure 4 shows the measured impedance spectra of the three tissues type. Does the data agree with previous findings? Would increasing the frequency past 10 kHz increase the difference between the tissue types? Is there a downside to using higher frequencies? Reference 15 used 86kHz. What would the impedance spectra for the tissues become if water content was varied? Would temperature of the tissue affect the impedance spectra in Figure 4? What temperature was used here? Figure 5a shows the real part of impedance during movement of the needle. Was this data recorded simultaneously with a moving needle, or was the needle held stationary in each tissue? Does movement or speed of the needle affect the measured impedance? What causes the variations in impedance, especially in the dermis and hypodermis region? The first and second hypodermis sections have different impedance values, why? What is the depth of each tissue? Figure 5b shows the mean values of the impedances, but is there any variation or deviation from these mean values during monitoring? How would the recorded data in Figures 4 and 5 change if measured in vivo? The authors indicate this method is for continuous monitoring. Would the impedance values vary for different patients and/or for different needles? Would this cause the operator to incorrectly identify the tissue type?

Reviewer 2 Report

1.This paper proposes an interesting method to assist with intra-articular injection therapy, but muscle texture or density varies widely in different parts of the body, the samples selected in this paper are not enough to support the topic.

2.The source of the experimental sample is not mentioned in this paper.

3.Why choose pork loin as experimental sample? The muscles at the joints are different from the pork loin.

4.What are the selection criteria for fresh pork loin for experiment? Why?

5.To obtain accurate experimental results, the experiments mentioned in this paper need to be repeated several times. In addition, the error bar should be added to the experimental results.

6. P2-3, The format of Equation 1-6 is incorrect.

Reviewer 3 Report

This is a well written paper. Description, methodology, experiments, results - all are presented clearly and with confidence. But I cannot recommend publishing the work in present form. Why? This work is in progress, still, not ready yet. Only the first part of a written scientific paper is given, but this is done correctly. 

Unfortunately, the using of lock-in measurement technology and needle electrodes for the detection of tissue type is known for years already. See the paper by Trebbels et al. in IEEE TBME, 2012, 59(2), also pay attention to the publications and press releases of the company Injeq OY, which produces similar devices.

The authors say that their device can improve intra-articular injection therapy. Please show this by performing the experiments in vitro, at least (better in vivo as well), and your paper will obtain the scientific and clinical value certainly high enough for publishing in the journal as SENSORS.

Round 2

Reviewer 1 Report

All of responses are good.

Reviewer 3 Report

The paper has been improved significantly. The list of references has been refined sufficiently to reflect the previous work of  earlier authors done in the same field in necessary extent. The paper can be published now as it is after amendments. However, please pay once more attention to the referring.